# Health-related quality of life and prevalence of six chronic diseases in homeless and housed people: a cross-sectional study in London and Birmingham, England

Dan Lewer, [1,2,3] Robert W Aldridge, [2,3] Dee Menezes, [2] Clare Sawyer, [4] Paola Zaninotto, [1] Martin Dedicoat, [5] Imtiaz Ahmed, [6] Serena Luchenski, [1,2,3] Andrew Hayward, [1,3] Alistair Story [3,4]

For numbered affiliations see end of article.

**Correspondence to**
Dan Lewer; d.lewer@ucl.ac.uk

## ABSTRACT

**Objectives** To compare health-related quality of life and prevalence of chronic diseases in housed and homeless populations.

**Design** Cross-sectional survey with an age-matched and sex-matched housed comparison group.

**Setting** Hostels, day centres and soup runs in London and Birmingham, England.

**Participants** Homeless participants were either sleeping rough or living in hostels and had a history of sleeping rough. The comparison group was drawn from the Health Survey for England. The study included 1336 homeless and 13 360 housed participants.

**Outcome measures** Chronic diseases were self-reported asthma, chronic obstructive pulmonary disease (COPD), epilepsy, heart problems, stroke and diabetes. Health-related quality of life was measured using EQ-5D-3L.

**Results** Housed participants in more deprived neighbourhoods were more likely to report disease. Homeless participants were substantially more likely than housed participants in the most deprived quintile to report all diseases except diabetes (which had similar prevalence in homeless participants and the most deprived housed group). For example, the prevalence of chronic obstructive pulmonary disease was 1.1% (95% CI 0.7% to 1.6%) in the least deprived housed quintile; 2.0% (95% CI 1.5% to 2.6%) in the most deprived housed quintile; and 14.0% (95% CI 12.2% to 16.0%) in the homeless group. Social gradients were also seen for problems in each EQ-5D-3L domain in the housed population, but homeless participants had similar likelihood of reporting problems as the most deprived housed group. The exception was problems related to anxiety, which were substantially more common in homeless people than any of the housed groups.

**Conclusions** While differences in health between housed socioeconomic groups can be described as a 'slope', differences in health between housed and homeless people are better understood as a 'cliff'.

## Strengths and limitations of this study

► The study is based on a large survey of 1336 homeless people.
► The survey includes data on outcomes that are rarely measured in homeless people, including long-term health conditions and health-related quality of life.
► An age-matched and sex-matched comparison group is used to compare the results with the general population.
► The cross sectional design means that causal pathways between homelessness and health are not investigated.
► The data were collected as part of a service evaluation and focus on a subset of diseases that cause substantial morbidity among homeless people.

## INTRODUCTION

Homelessness is an enduring social problem in high income countries.[1 2] It is associated with poor health outcomes, with cohort studies showing mortality risks of three to six times the general population.[3–7] The most common causes of death are usually (but not always[7]) accidents, drug overdoses, suicides and liver diseases. Homeless people also have substantially increased risk of death due to cardiovascular and respiratory diseases.[3 5 7 8]

There are few studies of morbidity among homeless people. Most have examined infections and mental health problems[9 10]; finding high relative and absolute frequencies. The few studies looking at broader outcomes have found that respiratory diseases, dental problems, headaches and skin diseases are also more common among homeless people than the general population.[11–15]

Inequalities in health in the general population have been widely observed. People in

BMJ

**Table 1**  Characteristics of study population

| | Find and treat (homeless) | Health Survey for England (housed) | P value |
|---|---|---|---|
| Sample size | 1336 | 13 360 | |
| **Age group (years)** | | | |
| 16–24 | 140 (10.5) | 1400 (10.5) | Matched |
| 25–34 | 364 (27.2) | 3640 (27.2) | |
| 35–44 | 374 (28.0) | 3740 (28.0) | |
| 45–54 | 315 (23.6) | 3150 (23.6) | |
| 55–64 | 143 (10.7) | 1430 (10.7) | |
| **Sex** | | | |
| Male | 249 (18.6) | 2490 (18.6) | Matched |
| Female | 1087 (81.4) | 10 870 (81.4) | |
| **Current smoker** | | | |
| Yes | 317 (23.7) | 3463 (25.9) | * |
| No | 115 (8.6) | 9839 (73.6) | |
| Missing | 904 (67.7) | 58 (0.4) | |
| **Drank alcohol every day last week** | | | |
| Yes | 101 (7.6) | 1087 (8.1) | * |
| No | 331 (24.8) | 12 271 (91.8) | |
| Missing | 904 (67.7) | 2 (0.01) | |
| **Injects drugs** | | | |
| Current | 82 (6.1) | – | * |
| Past | 191 (14.3) | – | |
| Never | 1034 (77.4) | – | |
| Missing | 29 (2.2) | 13 360 (100.0) | |
| **Prevalence of long-term conditions** | | | |
| Asthma | 244 (18.3) | 756 (5.7) | p<0.001 |
| COPD | 187 (14.0) | 180 (1.3) | p<0.001 |
| Epilepsy | 80 (6.0) | 108 (0.8) | p<0.001 |
| Heart problems | 103 (7.7) | 266 (2.0) | p<0.001 |
| Stroke | 24 (1.8) | 53 (0.4) | p<0.001 |
| Diabetes | 55 (4.1) | 431 (3.2) | p=0.098 |
| ***EQ-5D-3L (health-related quality of life)*** | | | |
| Reporting problems† | | | |
| Mobility | 251 (21.0) | 1192 (9.9) | p<0.001 |
| Self-care | 92 (7.7) | 377 (3.1) | p<0.001 |
| Usual activities | 178 (14.9) | 1261 (10.5) | p<0.001 |
| Pain | 432 (36.2) | 3034 (25.3) | p<0.001 |
| Anxiety | 646 (54.1) | 2229 (18.6) | p<0.001 |
| Visual analogue scale | | | |
| Median (IQR) | 65 (50–80) | 80 (70–90) | p<0.001 |
| Score<75 | 676 (63.8) | 1721 (29.3) | p<0.001 |

*Differences in health behaviours are not tested due to missing data.
†The proportions exclude 142 cases in the Find and treat group and 1345 cases in the Health Survey for England group who did not answer all EQ-5D-3L questions.
 COPD, chronic obstructive pulmonary disease; IQR, inter-quartile range.

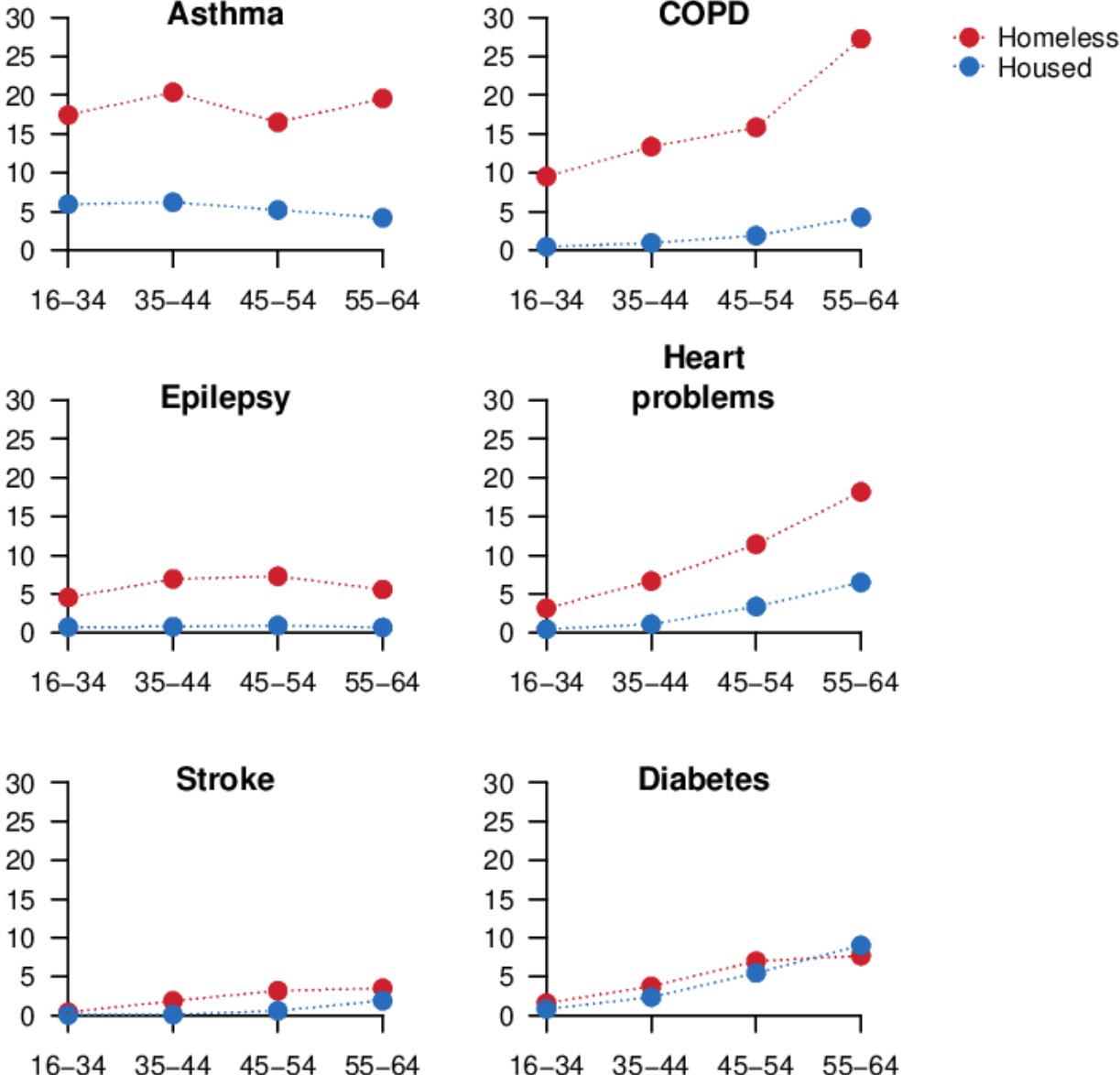

**Figure 1** Percentage reporting long-term conditions, by age group. COPD, chronic obstructive pulmonary disease.

lower socioeconomic groups have more risk factors and worse health outcomes.[16 17] Although poor health has been observed in homeless people, it is unclear whether homeless people have substantially worse health than the most deprived groups in the general population. We analysed data from a large survey of homeless people to understand how health-related quality of life and prevalence of chronic diseases in homeless people compares to different socioeconomic groups in the general population.

## METHODS
### Data sources
We used anonymised data from an evaluation of a mobile tuberculosis screening unit ('Find and Treat') that works alongside services for homeless people. Participants were people sleeping rough and accessing day centres and

soup kitchens, or people with a history of rough sleeping who are living in temporary hostel accommodations. Staff at the screening unit worked with homeless people to develop a health survey including questions on chronic diseases and health-related quality of life. Interviewer-led questionnaires were conducted in London from 1 July 2012 to 31 August 2012 and in Birmingham from 13 July 2014 to 18 July 2014 and from 16 March 2015 to 24 July 2015. We excluded 21 participants aged under 16 or over 64.

We selected a housed comparison group from the Health Survey for England. This is an annual series of cross-sectional surveys designed to be representative of adults living in private households. Participants are selected based on stratified random sampling of households across England. The full method is described elsewhere.[18] We used data from the 2006, 2008, 2010, 2011,

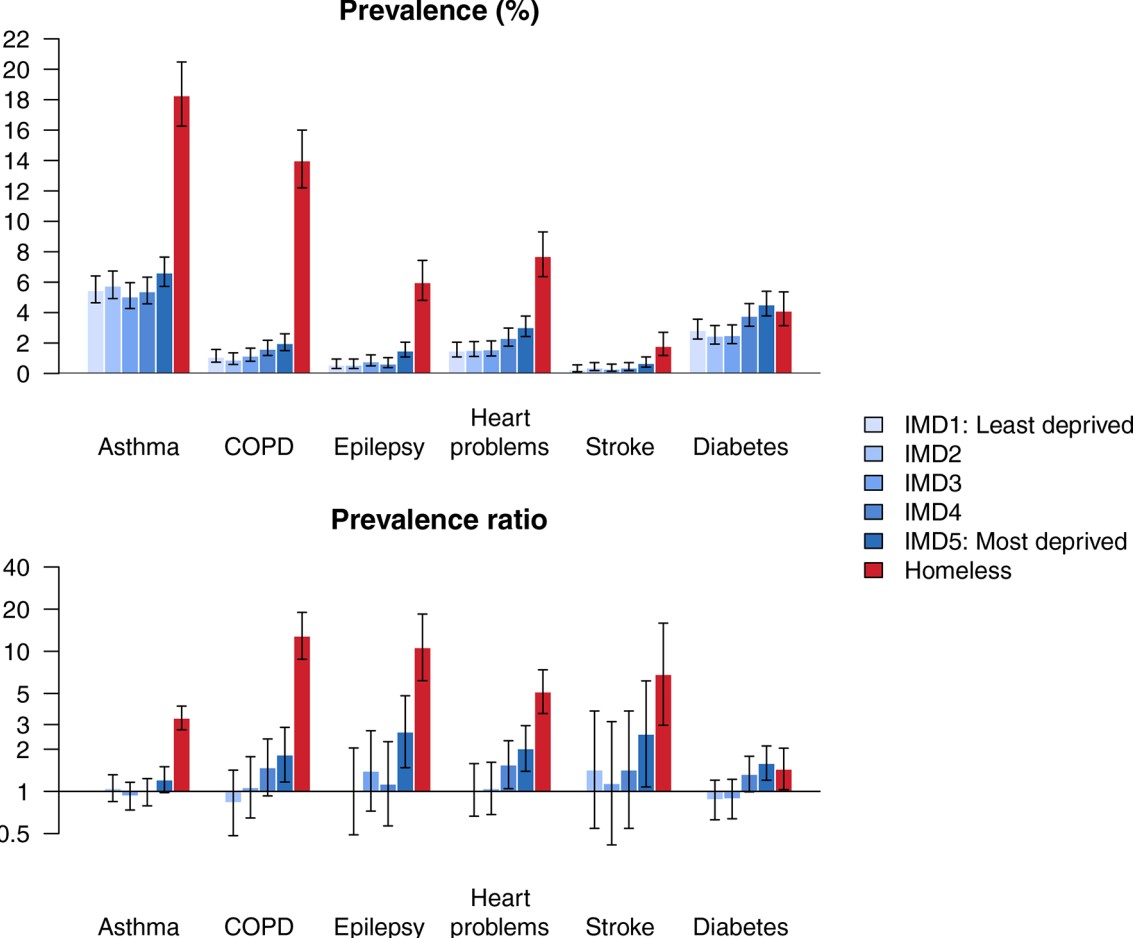

**Figure 2** Prevalence of long-term conditions (top panel) and prevalence ratios (bottom panel), with 95% CIs. COPD, chronic obstructive pulmonary disease; IMD, index of multiple deprivation.

2012 and 2014 surveys, limiting to participants from urban areas. Within each deprivation quintile, we stratified participants by age group and sex, and within each stratum selected a random sample of size double the number of homeless participants in the corresponding age and sex group. This produced an age-matched and sex-matched comparison group with ten housed participants for each homeless participant.

### Variables

The main outcomes are self-reported long-term conditions and health-related quality of life.

To establish prevalence of long-term conditions, homeless participants were asked whether a doctor or nurse had ever given a diagnosis of (1) asthma, (2) bronchitis, emphysema or obstructive airways disease (chronic obstructive pulmonary disease [COPD]), (3) epilepsy, (4) heart problems, including heart attack, angina, murmur or abnormal heart rhythms, (5) stroke and (6) diabetes. Housed participants were asked whether they had a health condition lasting or expected to last 12 months or more, with corresponding options of (1) asthma, (2) bronchitis, emphysema or other respiratory problems, (3) epilepsy, fits or convulsions, (4) heart attack, angina or other heart

problems, (5) stroke, cerebral haemorrhage or cerebral thrombosis and (6) diabetes including hyperglycaemia.

EQ-5D-3L[19] was used to measure health-related quality of life. Participants report no problems, some problems or extreme problems in five domains: mobility, pain, self-care, usual activities and anxiety. We created a binary variable showing whether participants reported 'none' versus 'some' or 'extreme' problems'. Participants also completed a visual-analogue scale (VAS) of overall health on the day of interview, where 0 is the worst imaginable state and 100 the best.

The main exposure was deprivation. The homeless group were classified as the most deprived, while the housed comparison group were classified using quintiles of the Index of Multiple Deprivation 2007.[20] This is an area-based index derived from levels of income, employment, health, disability, education, skills and training in small local areas of roughly 1500 residents.

Age was grouped into 16–24, 25–34, 35–44, 45–54 and 55–64, which was consistently available across datasets.

We also reported health behaviours: current smoking, drinking every day in the past week and currently injecting drugs.

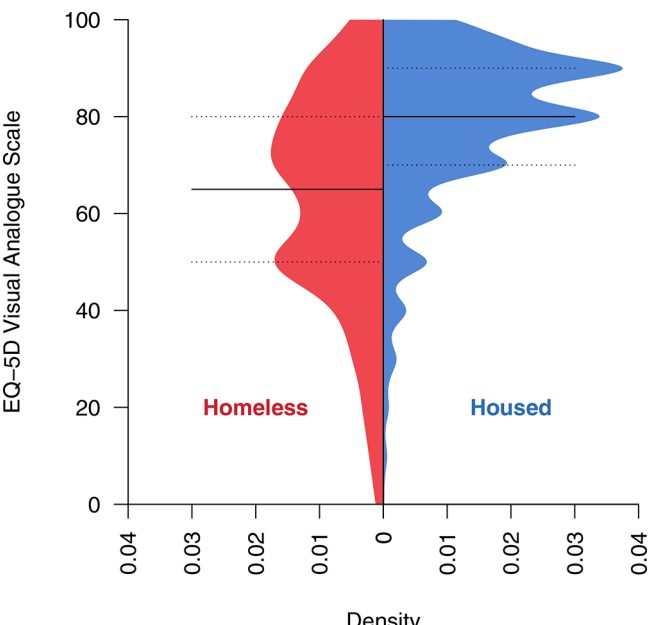

**Figure 3** Histograms of EQ-5D-Visual Analogue Scale score, by homeless/housed status. Horizontal lines show median and 25% and 75% quantiles.

## Statistical methods

We described homeless and housed participants by age group, sex, health behaviours, prevalence of long-term conditions and EQ-5D-3L characteristics. $\chi^2$ tests were used to test the difference between categorical variables and a Wilcoxon rank-sum test was used to test the difference between the EQ-5D-3L VAS score. We additionally calculated the prevalence of long-term conditions and EQ-5D-3L problems by deprivation group and calculated prevalence ratios with the least deprived housed group as the reference population, with Wald CIs. To test for social gradients in the housed population, we used $\chi^2$ test for trends with deprivation quintiles scored 1–5. We stratified prevalence of long-term conditions by age group and tested the association between age and long-term conditions using $\chi^2$ tests.

As a subanalysis of the effect of morbidity on inequalities in health-related quality of life, we used logistic regression to estimate the joint effect of deprivation and any morbidity (with the least deprived group with no long-term conditions as the reference group) on each EQ-5D-3L domain, adjusting for age group and sex. The results are shown in the online supplementary information.

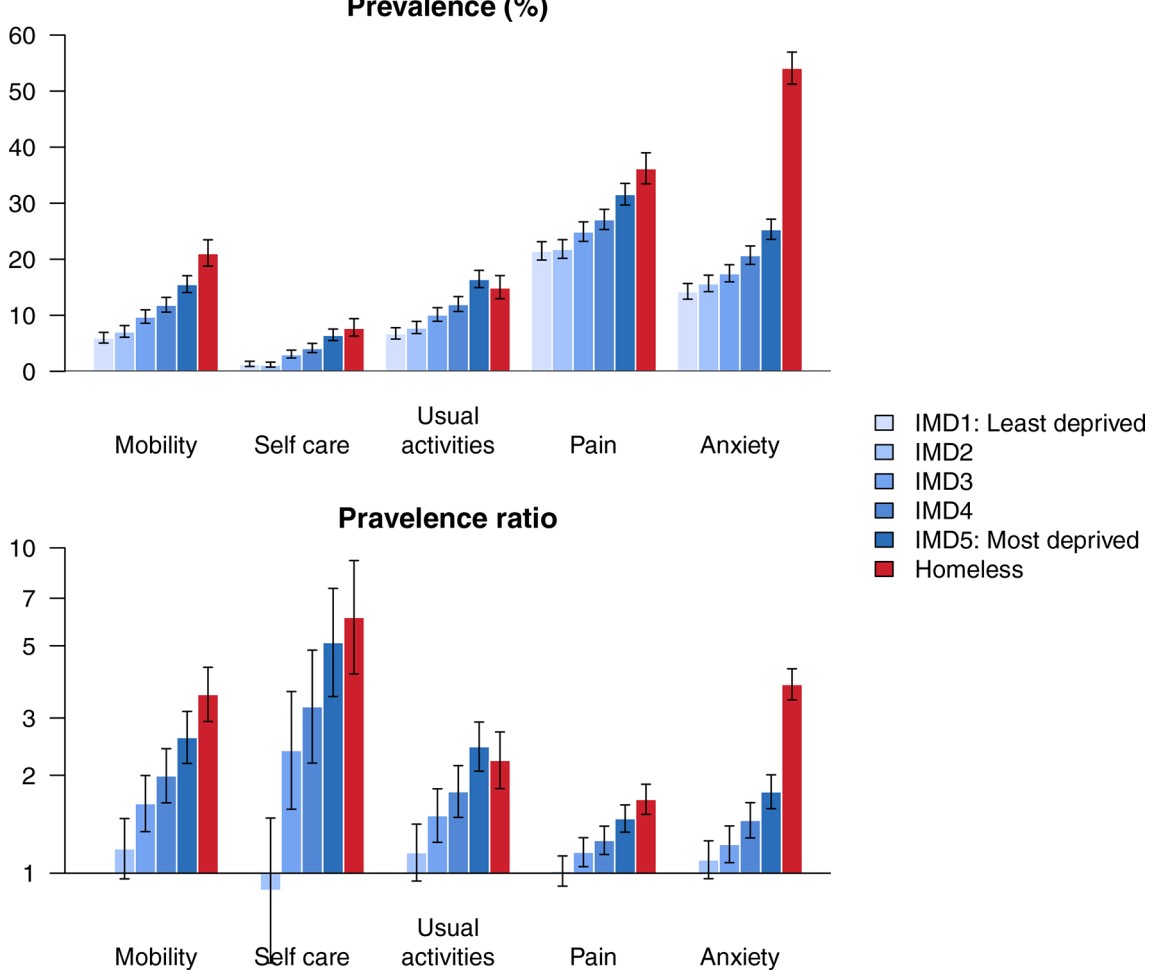

**Figure 4** Health-related quality of life: prevalence of problems by deprivation group (top panel) and prevalence ratios (bottom panel), with 95% CIs. IMD, index of multiple deprivation.

Some of the Health Survey for England data were collected several years before the surveys of homeless people, so we conducted a sensitivity analysis in which the procedures described above were repeated without the Health Survey for England years 2006 and 2008. These results are shown in online supplementary information.

Six participants in the homeless group did not provide their sex and were excluded. Age and long-term condition data were complete. 142/1336 (10.6%) homeless participants and 1353/13360 (10.1%) housed participants did not complete all EQ-5D-3L questions, with similar age ($\chi^2$ p=0.40) and sex ($\chi^2$ p=0.61) characteristics to those with complete data. These participants were excluded from analysis of EQ-5D-3L but retained for other analyses. Data on current smoking and everyday drinking were missing for 904/1336 (68%) homeless participants because these questions were only asked to participants in London. These data were missing for less than 1% of the housed group (table 1). Drug injection data were missing for 29/1336 (2.2%) homeless participants and all housed participants (as questions about drug injection are not included in the Health Survey for England). Data on health behaviours are shown in table 1 but associations with outcomes were not analysed.

All analyses were carried out using R V.3.4.2.

### Patient involvement
Clients of the Find and Treat service were involved in selecting the diseases included in this study, and in testing and refining the questionnaire.

### Ethics
This was an analysis of anonymised secondary data. We checked the requirement for ethical approval for this analysis using the National Health Service Health Research Authority/Medical Research Counsil approval checklist[21] and determined that ethical approval or additional informed consent was not required.

### RESULTS
#### Participants
The homeless sample consisted of 904 participants recruited from Birmingham and 432 from London (total=1336). The response rate was 77% in London and 71% in Birmingham. In the London sample, 73% reported current smoking and 23% reported drinking alcohol every day in the past week. The corresponding values in the housed group were 27% and 9%. Table 1 summarises the characteristics of the study population.

#### Long-term conditions
The prevalence of any long-term condition was 34.2% (95% CI 31.6% to 36.8%) in the homeless group and 12.1% (95% CI 11.5% to 12.6%) in the housed group. Asthma was the most commonly reported condition in both groups. All conditions except for diabetes were more common in the homeless group than in the housed

group. COPD, heart problems, stroke and diabetes were strongly associated with older age for both homeless and housed groups, while epilepsy and asthma were not (figure 1 and online supplementary information).

When we compared long-term conditions across deprivation quintiles in the housed population, we observed social gradients for all diseases except asthma (p<0.05), with participants living in deprived areas having higher risk (figure 2). For all conditions except diabetes, there were larger relative and absolute differences between the homeless group and all housed deprivation groups than there were between the housed deprivation groups. For example, the prevalence ratio comparing the risk of COPD in the most deprived quintile of housed participants versus the least deprived was 1.8 (95% CI 1.2 to 2.9), compared with 12.9 (95% CI 8.8 to 19.0) comparing the homeless group to the least deprived quintile.

### Health-related quality of life (EQ-5D-3L)
The homeless group reported worse overall health, with a median of 65 (IQR 50–80) on the EQ-5D-3L VAS, compared with median of 80 (IQR 70–90) for the housed group (figure 3).

802/1194 (67%) of the homeless group reported problems in at least one of the EQ-5D-3L domains (mobility, self-care, usual activities, pain and anxiety), compared with 4292/13360 (35.7%) of the housed group. The homeless group reported more problems than the housed group in all five domains.

When we compared EQ-5D-3L problems across deprivation quintiles in the housed population, we observed social gradients for all domains (p<0.0001), with participants living in deprived areas more likely to report problems (figure 4). In contrast to the results for disease prevalence, the homeless group and the most deprived housed quintile had similar likelihood of reporting problems related to mobility, self-care, usual activities and pain. For example, the prevalence of problems with usual activities in the most deprived quintile of housed participants was 2.5 times the least deprived group (95% CI 2.1 to 2.9), compared with 2.2 (95% CI 1.8 to 2.7) for the homeless group. However, the homeless group had substantially greater likelihood of reporting problems with anxiety than all housed groups. The most deprived quintile of housed participants was 1.8 times more likely to report problems with anxiety than the least deprived group (95% CI 1.6 to 2.0), compared with 3.8 (95% CI 3.4 to 4.2) for the homeless group.

The joint effects analysis (online supplementary information) of morbidity and deprivation showed that greater deprivation was associated with higher likelihood of reporting problems in all EQ-5D-3L domains, for participants both with and without long-term conditions. After accounting for morbidity, homeless participants had similar or lower likelihood of reporting problems in all domains except anxiety, when compared with than the most deprived housed participants.

## DISCUSSION
### Principle findings
Homeless people report substantially worse health than those in stable housing and are three times more likely to report a chronic disease. In particular, homeless people report higher prevalence of asthma, COPD, epilepsy and heart problems. Homeless people also report worse quality of life, and are more than twice as likely to report problems with anxiety. There are large differences between the homeless group and the most deprived housed group. When compared with the 'slopes' in health outcomes across deprivation quintiles, the inequalities in outcomes for homeless people appear more like a 'cliff'.

### Limitations
This is a cross-sectional study and does not provide insight into why homeless people have worse health than housed people. We aimed to describe the prevalence of health problems rather than understand these causal pathways, and therefore did not attempt to control for factors such as health behaviours.

EQ-5D-3L has not been validated in homeless people. However, it has been used in previous studies of the health of homeless people[22] and validity has been demonstrated in overlapping groups such as people who use illicit drugs.[23] There may be value in future research into the interpretation of quality of life instruments by homeless people.

### Comparisons with existing studies
The health of homeless people differs between countries and regions. However, the large inequalities between homeless and housed people are likely to persist. Studies that applied EQ-5D-3L to 73 homeless people in Italy[24] and 155 in Sweden[22] both found substantially more problems related to anxiety than other domains, reflecting our results. While there are few studies of long-term conditions in homeless people, existing studies have also found high frequency of non-communicable diseases.[11–15]

### Implications
There are many possible causes of the 'cliff' in long-term conditions and anxiety observed for homeless people, and the association between homelessness and poor health is most likely bidirectional. Poor health may precipitate homelessness if it leads to loss of income or breakdown of relationships. This is supported by data from Scotland showing high rates of health service use before first episodes of homelessness.[25] Homelessness is also likely to worsen physical health due to poor living conditions; use of tobacco, alcohol and drugs[11]; and poor access to health services.[26]

Despite the higher prevalence of long-term conditions, homeless people did not report substantially more quality-of-life problems (apart from problems related to anxiety). This may reflect a higher threshold for reporting problems, due to a low expectations of health or normalisation of pain and illness.

The age of onset of long-term conditions is related to the patterns observed across age groups (figure 1). The prevalence of asthma and epilepsy is flat across age groups in both the housed and homeless populations. These diseases typically onset in childhood and the higher prevalence in the homeless group may relate to circumstances in childhood or before homelessness. For example, heavy alcohol use and brain injury[27] are common precursors of homelessness and may contribute to higher prevalence of epilepsy. COPD and heart problems are strongly associated with adult risk factors, particularly smoking, which reflects the higher prevalence observed in older age groups, and the higher prevalence in the homeless group. The average age of first stroke is 71 (SD 13) in men in the UK,[28] which is older than the study population, reflecting the small number of cases and limited power to observe inequalities, though higher prevalence was still observed in the homeless group (table 1).

Research into the health of homeless people has focused on infections, mental health and 'external' causes such as drug overdose. Evaluations of interventions have also focused on these areas. For example, there is evidence that case management can improve the effectiveness of treatments for mental health problems, drug dependence and tuberculosis in homeless people.[29] Research into 'housing first', an approach that provides housing before engaging individuals in treatment for substance use or mental health problems, also looks at how it can improve outcomes in these areas.[30] The focus on these areas may have resulted from the high relative risks of disease,[7] in part due to their rarity in the general population, and because in most studies of mortality they account for the majority of deaths.

Our results show that respiratory diseases, epilepsy and heart problems are also common among homeless people in England. Homeless people encounter barriers to management of chronic diseases[31] and there is a need for research into effective healthcare. Case management should be further explored for management of chronic diseases. Existing evidence-based preventative interventions should also be adapted and evaluated for excluded groups. For example, smoking cessation interventions can reduce the risk of COPD and improve asthma symptoms, and should be tailored to homeless people and evaluated. There is also a need for routine surveillance data on the health of homeless people, to improve transparency and accountability for poor health in this population.

## CONCLUSION
There is a 'slope' in health outcomes across socioeconomic groups in the general population and a 'cliff' when we consider homeless people. These extreme differences extend to long-term physical health conditions as well as infections and mental health.

**Author affiliations**
[1]Institute of Epidemiology and Healthcare, University College London, London, UK

[2]Institute of Health Informatics, University College London, London, UK
[3]Collaborative Centre for Inclusion Health, University College London, London, UK
[4]Find & Treat, University College London Hospitals NHS Foundation Trust, London, London, UK
[5]Infectious Diseases and Tropical Medicine, University Hospitals Birmingham NHS Foundation Trust, Birmingham, UK
[6]Respiratory Medicine, Sandwell and West Birmingham Hospitals NHS Trust, Birmingham, UK

**Contributors** AS and AH conceived of the initial idea. DL, RWA, DM, CS, PZ, MD, IA, SL, AH and AS contributed to the design of the study. DL conducted data analysis and wrote the first draft of the paper. DL, RWA, DM, CS, PZ, MD, IA, SL, AH and AS contributed to the main content of the manuscript, provided comments on the final draft and approved the manuscript before submission.

**Funding** DL is funded by an NIHR Doctoral Research Fellowship (DRF-2018-11-ST2- 016). RWA is funded by a Wellcome Trust Clinical Research Career Development Fellowship (206602/Z/17/Z). IA is funded by Sandwell Hospital and Public Health England. SL is funded by an NIHR Clinical Doctoral Research Fellowship (ICA-CDRF-2016-02-042).

**Disclaimer** Professor Andrew Hayward is a National Institute for Health Research (NIHR) Senior Investigator. The views expressed in this article are those of the author(s) and not necessarily those of theNIHR, or the Department of Health and Social Care.

**Competing interests** AH is Trustee of the UK-based charity 'Pathway (healthcare for homeless people)'. AS is Clinical Lead for the Find & Treat Service; data were collected from homeless clients of this service.

**Patient consent for publication** Not required.

**Provenance and peer review** Not commissioned; externally peer reviewed.

**Data sharing statement** Data from the survey of homeless people have not been made publicly available because the some individuals may be identifiable. Data from the Health Survey for England used in this study are available via the UK Data Service, serial numbers SN6397, SN6986, SN7260, SN7480, SN7919.

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
