## [Reviewer comments · BMJ Open]

ARTICLE DETAILS

TITLE (PROVISIONAL)	Health-related quality of life and prevalence of six chronic diseases in homeless and housed people: a cross-sectional study in London and Birmingham, England
AUTHORS	Lewer, Dan; Aldridge, Robert; Menezes, Dee; Sawyer, Clare; Zaninotto, Paola; Dediccoat, Martin; Ahmed, Imtiaz; Luchenski, Serena; Hayward, Andrew; Story, Alistair

VERSION 1 - REVIEW

REVIEWER	Stewart Mercer General Practice and Primary Care University of Glasgow Scotland
REVIEW RETURNED	10-Sep-2018

GENERAL COMMENTS	This is a very interesting paper, and provides useful data on a very hard to reach group. It adds significantly to the literature. I do think this paper is worthy of publication, but I have suggested changes which probably qualify as major rather than minor. Firstly, the authors state that no ethical approval was required as it was part of a service (TB Screening). However, they also say that not only those being screened were eligible to take part. Was the need for ethical approval actually discussed with an ethics committee? Or screened on the online NREC system? This needs to be clarified in the text. Secondly, the control group was drawn from census data where the % of mean was well below 50%, whereas the homeless group was predominately male. This makes comparison difficult, and it might be better to simply look at males. I think a statistical review regarding how best to adjust for gender differences if all the data is used is also warranted. I am also concerned that the deprivation gradient examined was quintiles of deprivation. There are often big differences between deciles 9 and 10 (most deprived) and I would like to see all the analysis and the graphs re-drawn using deciles rather than quintiles.
---

REVIEWER	Kelly J Kelleher. MD, MPH Nationwide Children's Hospital
REVIEW RETURNED	26-Sep-2018

GENERAL COMMENTS

This manuscript presents the results of a large survey from two cities of persons stably housed and corresponding group of homeless individuals. The topic is an urgent public health issue with growing numbers of homeless or housing insecure individuals in major cities around the world because of unaffordable housing.

The authors provide an important service in examining the prevalence of chronic diseases and illnesses not often considered in the research on the health of homeless individuals. They correctly note that prior research has attended primarily to mental disorders and infections. In addition, they engage a large group of survey respondents so that they can classify the comparison group by class/income and provide more in-depth analysis of the gradient of health status across the chronic conditions.

Several other strengths are noted. The authors include an important quality of life measure and diverse population from many locations. The language and communication are clear, and they do not exceed their findings in discussing the results. Best of all, the concept of a health 'cliff' for homeless individuals versus 'gradient' seen among housed populations is a welcome addition to the literature.

Some changes would strengthen the manuscript. First, the authors inadequately define the homeless population. Although they note that all of their sample of homeless individuals were gathered at soup runs and shelters and related sites next to a tuberculosis screening program for the homeless, they do not say whether any of these services were used by people with housing. Relatedly, it is not clear if all persons were street-living or whether they may have been housing insecure such that they were sleeping temporarily with friends, neighbors or relatives.

Secondly, the authors acknowledge the importance of chronic disease but do not include drug use, HIV, hepatitis or skin diseases, all highly prevalent in the homeless in their findings. It is not clear if the chronic illnesses considered are comorbid or independent from these other epidemics in the homeless individuals.

A minor point is that the manuscript suggests that 'research ethics is not needed', but the more accurate statement would be that 'informed consent was not needed' because no identifiable human subjects information was employed.

Also, from a clinical perspective it is unclear whether these respondents suffered from anxiety directly or PTSD/trauma. Street living is associated with high rates of victimization and violence exposure and some inference to that would be useful for the reader, or even better, more information on violence exposure or trauma.

Finally, the authors employ a cross sectional survey, the only feasible method of collecting this scale of data. While useful, the manuscript repeatedly hints that homelessness 'causes' or is directionally related to the onset of these conditions or at the very least the worsening of them. However, it is also likely that these conditions may have precipitated homelessness in at least some individuals because those persons would have been unable to make rent. Some acknowledgment of the bidirectionality is

	probably important and careful review to not point to causation in the discussion is indicated.
--	---

REVIEWER	Sara Conti Research Center on Public Health Department of Medicine and Surgery University of Milano-Bicocca Monza (Italy)
REVIEW RETURNED	01-Dec-2018

GENERAL COMMENTS	The paper addresses health inequalities in homeless people aged 16-64 as compared to housed people of the same age class in UK, but it also stratifies the latter according to quintiles of a small-area deprivation index, available from UK 2011 Census. Endpoints are both chronic conditions and self-reported quality of life. The analysis is interesting, as it reveals that the prevalence of chronic conditions among homeless people is consistently higher than that observed among housed ones, when focusing on housed people in the lowest quintile of deprivation index. On the other hand, perceived quality of life among homeless people is different from that perceived from housed people in the lowest quintile of deprivation index only in terms of anxiety and depression and VAS scale. The paper is overall well written and clear, but I have some minor comments that I feel might improve the paper. 1) Page 4, Lines 8-14. Data from the Health Survey for England are from 2008, 2010, 2011, 2012 and 2014, while homeless people were interviewed in 2012, 2014 and 2015. Given that the study is considered to be cross-sectional, why did authors choose to include data from 2008 and 2010, also considering the global financial crisis of 2007-2008? 2) Methods section. Authors should provide more details regarding the analysis of the VAS: did they test for normality? If they did not, I would suggest to check for it, because if such condition is not satisfied the analysis on VAS (especially figure 3) should be reconsidered by reporting median and their confidence intervals. 3) Page 5, Lines 47-53. I would suggest to report p-values of comparisons between homeless and housed people as for mean age and sex. Age should be tested for normality, following the same considerations reported for the VAS. 4) Table 1. I would suggest to report p-values of the comparisons between the homeless and the housed group for all variables listed in the tables, as this would allow the reader to immediately perceive the differences between these two groups. In the section regarding "Country of birth" of housed people the number of subject in each group is missing. I would suggest to also report median and interquartile range for the VAS, for the reason reported in comment 2). The difference should be tested through Wilcoxon test. 5) Page 7, Lines 46-55, and page 8, Lines 1-4. Unfortunately, I disagree with this paragraph concerning the limitation due to shorter survival of homeless people as compared to housed ones. The population alive at the moment of the survey should reflect the age- and gender- and health-structure of the average population of each group at each comparable moment in time: if the population is stable over time, people that die young because they have a worse health conditions are replaced by similar people. Indeed the population is dynamic, and not fixed as for cohort studies.
--

	6) Page 8, Lines 35-40. The result of the analysis restricted to people with a chronic condition are interesting and should be also reported in the results section. 7) Overall, there are some references to figures and tables that should be checked: sometimes the number of the figure or table is missing.
--	--

REVIEWER	Martin Siegel Technische Universität Berlin, Dept. of Empirical Health Economics, Berlin, Germany
REVIEW RETURNED	06-Dec-2018

GENERAL COMMENTS	Review on the paper “Slopes and cliffs in health inequalities: a cross-sectional study comparing the health of homeless and housed populations of England”: Overall, the paper raises an interesting question and addresses the understudied group of homeless people. The authors find a gradient among different regional deprivation quantiles and a gap between housed and homeless peoples’ health. As only a statistical review was requested, I will focus on statistical issues and the respective conclusions: A description of how the housed comparison group was selected would be required. How was that sampled? Were observations matched e.g. by risk of homelessness? By something else? Or is it just a random sample from urban areas all over England or UK? Or from the two cities where the study was conducted? How certain is it that there are no repeated measurements of the same individuals counted as different individuals? I.e. is the survey a panel study or are the data resampled? The authors state that they used direct standardization for the prevalence rates. While this is a common and adequate approach, the authors should elaborate on how they did that and provide a reference, since there are different approaches out there. I do not understand the reference group statement: If the direct standardization was computed separately for each deprivation group, what is the reference group good for? “Standard errors were calculated using the method described by the Centres for Disease Control and Prevention” - what does that mean? Please elaborate what you did exactly. The Index of Multiple Deprivation is a nice approach to combine different deprivation domains into one univariate score. While I appreciate the use of the IMD, the health and disability domains should be excluded when looking at health as an outcome. It would be a tautological approach otherwise, explaining health with itself. The country of origin does not seem to re-appear in the results or discussion. Why use an approximation from the 2011 census at risk of ecological fallacy, if the information is not further exploited? A number of statistical strategies are mixed: Direct standardization, sample stratification, logistic regression using
--

	control variables, chi-squared tests of contingency tables. The whole methods description is confusing, no rationales are provided for choosing strategies (why mix standardization with logistic regression and sample stratification?). Although being primary data collected for this study, the data on homeless people do not contain information about health behavior (the term lifestyle for addiction disorders appears to be cynical at best to me). The statement on cross-sectional designs and deceased homeless people is again a bit confusing. I reckon that this will rather change the age structure than lets us underestimate the burden of disease. This should be covered by the standardization procedure. The implications section is highly speculative. No health behavior is surveyed, but now the authors speculate that it is the homeless peoples' behavior that makes them sick. They have good reasons not to include lifestyle factors, but then they should avoid the speculative discussion. The statement on the potentially good or even better coping strategies among homeless people seems problematic. Boldly speaking: If they had better coping strategies, why would they be homeless? There is some research in the Journal of the Royal Statistical Society (Series B) on anchoring vignettes showing that subjective health outcomes such as EQ5D may be considerably affected by different notions of poor and good health, or living with or without problems. The other explanation, and this is completely missing, is that the homeless have a very different notion of good health and living without problems, using their usual health status as a benchmark (again boldly speaking: People get used to suffering). The conclusion again focuses on health behavior, which is not sound in a study which didn't include behavior in their analysis.
--	--

VERSION 1 – AUTHOR RESPONSE

Reviewer(s)' Comments to Author:

Reviewer: 1

5. Firstly, the authors state that no ethical approval was required as it was part of a service (TB Screening). However, they also say that not only those being screened were eligible to take part. Was the need for ethical approval actually discussed with an ethics committee? Or screened on the online NREC system? This needs to be clarified in the text.

** Response: we have clarified in the text that this is a secondary analysis of anonymised data and the need for ethical approval was screened using the NREC checklist.

6. Secondly, the control group was drawn from census data where the % of mean was well below 50%, whereas the homeless group was predominately male. This makes comparison difficult, and it might be better to simply look at males. I think a statistical review regarding how best to adjust for gender differences if all the data is used is also warranted.

** Response: We agree with the referee that the original method was making the comparison difficult. We have therefore revised the methods with a statistician (now included in the author list) and decided to use a matching approach to select the comparison (housed) group. As explained in the methods we have selected the comparison group based on the sex and age profile of the homeless group. This approach allows a more direct comparison between the homeless and housed groups.

7. I am also concerned that the deprivation gradient examined was quintiles of deprivation. There are often big differences between deciles 9 and 10 (most deprived) and I would like to see all the analysis and the graphs re-drawn using deciles rather than quintiles.

** Response: We agree that this would be a useful analysis, but the Health Survey for England data only includes quintiles of deprivation, and does not include the participant's LSOA of residence (which would allow you to look up the deciles).

Reviewer: 2

8. First, the authors inadequately define the homeless population. Although they note that all of their sample of homeless individuals were gathered at soup runs and shelters and related sites next to a tuberculosis screening program for the homeless, they do not say whether any of these services were used by people with housing. Relatedly, it is not clear if all persons were street-living or whether they may have been housing insecure such that they were sleeping temporarily with friends, neighbors or relatives.

** Response: Sorry that this information was omitted. We have now clarified that all participants were either sleeping rough or were living in hostels and had a history of sleeping rough.

9. Secondly, the authors acknowledge the importance of chronic disease but do not include drug use, HIV, hepatitis or skin diseases, all highly prevalent in the homeless in their findings. It is not clear if the chronic illnesses considered are comorbid or independent from these other epidemics in the homeless individuals.

** Response: we have now included data on health behaviours (table 1), which shows the relatively high prevalence of injecting drugs (20% currently or previously injected). The health survey focused on diseases that have received limited attention in this population to date, and we did not have data on skin diseases, HIV and hepatitis (which, as the reviewer notes, have already been shown to be important causes of morbidity). Given the focus on a subset of chronic diseases, we decided not to analyse co-morbidity or multimorbidity.

10. A minor point is that the manuscript suggests that 'research ethics is not needed', but the more accurate statement would be that 'informed consent was not needed' because no identifiable human subjects information was employed.

** Response: we have updated our statement on ethics to clarify that the study is an analysis of anonymised secondary data.

11. Also, from a clinical perspective it is unclear whether these respondents suffered from anxiety directly or PTSD/trauma. Street living is associated with high rates of victimization and violence exposure and some inference to that would be useful for the reader, or even better, more information on violence exposure or trauma.

** Response: We agree with the statement and it matches up with our own experience. Unfortunately, we did not have data on anxiety (apart from the EQ5D domain), PTSD or trauma, and therefore we could not include these conditions in the study.

12. Finally, the authors employ a cross sectional survey, the only feasible method of collecting this scale of data. While useful, the manuscript repeatedly hints that homelessness 'causes' or is directionally related to the onset of these conditions or at the very least the worsening of them. However, it is also likely that these conditions may have precipitated homelessness in at least some individuals because those persons would have been unable to make rent. Some acknowledgment of the bidirectionality is probably important and careful review to not point to causation in the discussion is indicated.

** Response: We agree with the referee that causation cannot be implied from cross-sectional data. We have revised the text to recognise the limitations of the cross-sectional method and the bidirectionality between homelessness and health. We have also now included some data on health behaviours. While this does not provide evidence of causation, it supports a discussion that tobacco, alcohol and drugs may play a role in the poor health outcomes of homeless people.

Reviewer: 3

13. Page 4, Lines 8-14. Data from the Health Survey for England are from 2008, 2010, 2011, 2012 and 2014, while homeless people were interviewed in 2012, 2014 and 2015. Given that the study is considered to be cross-sectional, why did authors choose to include data from 2008 and 2010, also considering the global financial crisis of 2007-2008?

** Response: We chose to use these Health Survey for England years to give the study sufficient power. We have now used a matching approach rather than standardisation, which also required us to use data from 2006 so that a 2:1 ratio of housed:homeless within each deprivation quintile could be achieved. We recognise that the difference in period may limit the comparability between the homeless and housed populations, and we therefore conducted a sensitivity analysis in which the data from 2006 and 2008 are excluded (shown in the supplementary material), showing that the results do not change materially when these years are excluded. We also used logistic regression to test for a trend in the prevalence of each chronic disease over time in the Health Survey for England and found no evidence of a change. (results available on request).

14. Methods section. Authors should provide more details regarding the analysis of the VAS: did they test for normality? If they did not, I would suggest to check for it, because if such condition is not satisfied the analysis on VAS (especially figure 3) should be reconsidered by reporting median and their confidence intervals.

** Response: Thank you for this comment. As the referee suggested VAS was non-normally distributed, and in this revised version of the paper we report medians and IQRs, and have also included a histogram/density plot (figure 3) to show the distribution of values.

15. Page 5, Lines 47-53. I would suggest to report p-values of comparisons between homeless and housed people as for mean age and sex. Age should be tested for normality, following the same considerations reported for the VAS.

** Response: To improve the comparison with the housed group we have adopted a matching approach so the housed and homeless group have the same age and sex profile. We have included p-values in Table 1 as suggested by the referee. Age is included as a categorical variable.

16. Table 1. I would suggest to report p-values of the comparisons between the homeless and the housed group for all variables listed in the tables, as this would allow the reader to immediately perceive the differences between these two groups. In the section regarding "Country of birth" of housed people the number of subject in each group is missing. I would suggest to also report median

and interquartile range for the VAS, for the reason reported in comment 2). The difference should be tested through Wilcoxon test.

** Response: we have now used a matching approach so direct comparisons between the homeless and housed groups are possible. We have included p-values comparing the frequency of all outcomes. We have decided not to report the country of birth data, due to the limitations of using Census data as a proxy for the Health Survey for England.

17. Page 7, Lines 46-55, and page 8, Lines 1-4. Unfortunately, I disagree with this paragraph concerning the limitation due to shorter survival of homeless people as compared to housed ones. The population alive at the moment of the survey should reflect the age- and gender- and health-structure of the average population of each group at each comparable moment in time: if the population is stable over time, people that die young because they have a worse health conditions are replaced by similar people. Indeed the population is dynamic, and not fixed as for cohort studies.

** Response: we accept this, and have removed this part of the discussion.

18. Page 8, Lines 35-40. The result of the analysis restricted to people with a chronic condition are interesting and should be also reported in the results section.

** Response. Thank you for your positive comment. We did consider bringing the results into the main article, but unfortunately we were restricted by space.

19. Overall, there are some references to figures and tables that should be checked: sometimes the number of the figure or table is missing.

** Response: we have now checked carefully through the tables and figures.

Reviewer: 4

20. A description of how the housed comparison group was selected would be required. How was that sampled? Were observations matched e.g. by risk of homelessness? By something else? Or is it just a random sample from urban areas all over England or UK? Or from the two cities where the study was conducted? How certain is it that there are no repeated measurements of the same individuals counted as different individuals? I.e. is the survey a panel study or are the data resampled?

** Response: We have included a brief description of the method and a reference to a more detailed description. Briefly, the Health Survey for England is a cross-sectional survey that uses a new random sample each year from the whole of England (urban and rural). The same people are not selected again in future surveys. We limited to participants from urban areas and matched them to the homeless sample based on age and sex profiles.

21. The authors state that they used direct standardization for the prevalence rates. While this is a common and adequate approach, the authors should elaborate on how they did that and provide a reference, since there are different approaches out there. I do not understand the reference group statement: If the direct standardization was computed separately for each deprivation group, what is the reference group good for?

** Response: We agree with the referee that the standardisation method was probably not giving the best comparison, we therefore revised the method and used a matching approach to select the comparison group based on the age and sex profile.

22. "Standard errors were calculated using the method described by the Centres for Disease Control and Prevention" - what does that mean? Please elaborate what you did exactly.

** Response: This method is no longer used in the revised version of the paper.

23. The Index of Multiple Deprivation is a nice approach to combine different deprivation domains into one univariate score. While I appreciate the use of the IMD, the health and disability domains should be excluded when looking at health as an outcome. It would be a tautological approach otherwise, explaining health with itself.

** Response: We agree with this principle. Unfortunately, Health Survey for England only provides the overall IMD quintile of each participant and does not include LSOA (which would allow us to lookup individual IMD domain scores). Although this is a limitation, the effect on the results should be small because the IMD score excluding the health and disability domain is similar to the overall IMD score. This is partly because the health and disability domain only has 13.5% weight, and partly because it is correlated with other domains. We used the IMD2015 published data to check the differences between the overall score and the score excluding the health and disability domain. 90% of LSOAs are in the same quintile, and the remaining 10% are one quintile high or lower.

24. The country of origin does not seem to re-appear in the results or discussion. Why use an approximation from the 2011 census at risk of ecological fallacy, if the information is not further exploited?

** Response: we have excluded this data from the table, given the limitations of using Census data as a proxy.

25. A number of statistical strategies are mixed: Direct standardization, sample stratification, logistic regression using control variables, chi-squared tests of contingency tables. The whole methods description is confusing, no rationales are provided for choosing strategies (why mix standardization with logistic regression and sample stratification?).

** Response: We agree with the referee and in order to address this comment we have now used a different approach for the selection of the comparison group. Furthermore we have revised the method section.

26. Although being primary data collected for this study, the data on homeless people do not contain information about health behavior (the term lifestyle for addiction disorders appears to be cynical at best to me).

** Response: the survey of homeless people did include some data on health behaviours (smoking, everyday drinking and injecting drugs). We originally did not include this data because the smoking and drinking questions were not asked to participants from Birmingham. Following the referee's comment, we now include this data for the London sample. We have used the term 'health behaviours' rather than 'lifestyle'.

27. The statement on cross-sectional designs and deceased homeless people is again a bit confusing. I reckon that this will rather change the age structure than lets us underestimate the burden of disease. This should be covered by the standardization procedure.

** Response: we accept this, and have removed this part of the discussion.

28. The implications section is highly speculative. No health behavior is surveyed, but now the authors speculate that it is the homeless peoples' behavior that makes them sick. They have good reasons not to include lifestyle factors, but then they should avoid the speculative discussion.

** Response: we have revised the discussion to highlight the limitations of the design. We have also added data on health behaviours to support this part of the discussion.

27. The statement on the potentially good or even better coping strategies among homeless people seems problematic. Boldly speaking: If they had better coping strategies, why would they be homeless? There is some research in the Journal of the Royal Statistical Society (Series B) on anchoring vignettes showing that subjective health outcomes such as EQ5D may be considerably affected by different notions of poor and good health, or living with or without problems. The other explanation, and this is completely missing, is that the homeless have a very different notion of good health and living without problems, using their usual health status as a benchmark (again boldly speaking: People get used to suffering).

** Response: we agree with this comment and have revised the discussion accordingly.

28. The conclusion again focuses on health behavior, which is not sound in a study which didn't include behavior in their analysis.

** Response: we have revised the conclusion so it is focused on the findings of the study (the difference in health outcomes that we measured).

VERSION 2 – REVIEW

REVIEWER	Stewart Mercer University of Glasgow, Scotland
REVIEW RETURNED	04-Feb-2019

GENERAL COMMENTS	The authors have responded satisfactorily to the queries I raised when I reviewed the first submission, and I am not happy to recommend publication.
--

REVIEWER	Kelly Kelleher Nationwide Children's Hospital
REVIEW RETURNED	08-Feb-2019

GENERAL COMMENTS	This revised manuscript compares survey responses from homeless individuals from those in various economic classes who are housed over several years in London and Birmingham. The authors seek to provide new information on rates of chronic medical conditions not usually considered among the homeless as well as to provide a much larger sample of homeless individuals with comparison on a quality of life measure. As such, this is important information. Besides attending to an important problem, many of the previously noted strengths were retained in this version. The authors include a large sample, administer a QOL tool that is widely used for comparability, engaged homeless individuals in design of the survey, and prepared detailed analyses with various checks for sensitivity. In addition, this version has significant improvements from the previous draft with much less commentary in the introduction and discussion. The inclusion of detailed information on drop outs and the comparison for a temporal trend were both useful. The authors also inserted a discussion about dual direction causality which makes sense for anyone working with this population.
--

There are other limitations, but these limitations are largely a result of the original surveys and their comparability, not of any study or writing issues. Most of these are covered by the authors in the limitations, which were shortened.

The figures are especially useful in underlining the severity of the problem.

This revised manuscript compares survey responses from homeless individuals from those in various economic classes who are housed over several years in London and Birmingham. The authors seek to provide new information on rates of chronic medical conditions not usually considered among the homeless as well as to provide a much larger sample of homeless individuals with comparison on a quality of life measure. As such, this is important information.

Besides attending to an important problem, many of the previously noted strengths were retained in this version. The authors include a large sample, administer a QOL tool that is widely used for comparability, engaged homeless individuals in design of the survey, and prepared detailed analyses with various checks for sensitivity.

In addition, this version has significant improvements from the previous draft with much less commentary in the introduction and discussion. The inclusion of detailed information on drop outs and the comparison for a temporal trend were both useful. The authors also inserted a discussion about dual direction causality which makes sense for anyone working with this population.

There are other limitations, but these limitations are largely a result of the original surveys and their comparability, not of any study or writing issues. Most of these are covered by the authors in the limitations, which were shortened.

The figures are especially useful in underlining the severity of the problem.